# A Novel EWOD Platform for Freely Transporting Droplets in Double and Single-Plate Structures

**DOI:** 10.3390/mi15060797

**Published:** 2024-06-17

**Authors:** Yii-Nuoh Chang, Ting-Rui Huang, Da-Jeng Yao

**Affiliations:** 1Institute of Nano Engineering and MicroSystems, College of Engineering, National Tsing Hua University, Hsinchu 30013, Taiwan; enochchang070@gmail.com; 2Department of Power Mechanical Engineering, College of Engineering, National Tsing Hua University, Hsinchu 30013, Taiwan; terrence0929@gmail.com

**Keywords:** EWOD, droplet manipulation, curved channel

## Abstract

This study developed a novel dielectric wetting microfluidic operation platform combining parallel-plate and coplanar-plate regions with a curved surface structure as the connection structure. With the new electrowetting on dielectric (EWOD) platform, “droplet pull-out” has been successfully achieved and viewed as an essential new operation for microfluidics with the dielectric wetting technique. The EWOD system is divided into a PDMS substrate top plate and an indium tin oxide (ITO) glass substrate as a bottom layer on this chip. In the parallel-plate region, the droplets can be generated and transported through the square parallel electrodes; in the single-plate area, the droplets can be pulled out from the parallel structure, transported and mixed through the common grounded coplanar electrodes. In dielectric wetting performance testing, coplanar electrodes can apply a maximum driving force of 31.22 µN to DI water and 13.38 µN to propylene carbonate (PC). This driving force is sufficient to detach the sample from the top cover and pull the sub-droplet from the parallel plate structure for DI water, PC and polyethylene glycol diacrylate (PEGDA) buffer. The novel EWOD system also possesses the advantage of precise volume control for liquid samples; the volume error of the generated droplet can be controlled within 0.1% to 2%.

## 1. Introduction

The microfluidic system is a research field closely related to techniques such as microelectromechanical chip design, optical display and molecular biology applications. These techniques have made microfluidics an independent science and promoted its development with fabrication methods or application materials. Since the EWOD system was published, a small amount of required reagent buffer, precise volume control, and fast reaction time have made this technique widely used in biomedical research. The parts of the EWOD system are mainly divided into the top plate, middle oil fluid layer, and bottom chip. With the design of a digital microfluidic circuit, CJ Kim’s team accomplished four fundamental droplet operations: creation, transporting, cutting and merging at low operation voltage [1,2,3,4,5]. For EWOD technique applications, the parallel-plate system exhibits excellent droplet volume control performance with four fundamental operations [6]. The double-plate aspect ensures that droplets are shielded from physical interferences and evaporation by being surrounded by an immiscible fluid, enhancing stability and reliability. In comparison, the coplanar EWOD system cannot efficiently perform creating or cutting operations. Despite the difficulty in droplet creation and cutting, the coplanar system still offers a few advantages, such as promoting system integration with different lab tools, improving the biomedical samples sticking issue, and achieving more efficient liquid mixing [7,8].

Although two types of EWOD systems have their advantages, it is hard to combine two structures since there is a boundary resistance at the capillary structure’s edge, making it challenging for the droplet to pull out from the parallel structure. In 2022, a top plate design was proposed by Chang et al. with a curved surface structure to overcome the top plate boundary resistance, pulling out the droplets from the capillary structure; there is room for further improvement in the EWOD system structure and the fabrication process stability of curve surface top plate [9]. Based upon previous research on the curved surface top plate, this study integrates this structure into the EWOD system and hopes to develop a novel EWOD platform that combines parallel-plate and single-plate EWOD configurations. The novel platform can provide the minimized biomolecule adsorption, and broad sensing capabilities can significantly enhance drug discovery processes, where maintaining sample integrity is critical. The platform’s stability and reduced evaporation make it ideal for precise and reliable diagnostic assays requiring consistent conditions over extended periods. The platform is tested with a few liquid materials such as DI water, PC, and PEGDA to assess its performance in volume control and liquid sample manipulation.

## 2. Materials and Methods

### 2.1. Design and Fabrication Methods of EWOD Chip

The dielectric wetting wafer used in this experiment was mainly designed with Auto CAD 2018 software. A 6 × 5 electrode array composed of coplanar electrodes in the chip’s center mainly works as the single-plate open area, and parallel-plate operating areas are designed along the four sides, presenting a pinwheel-like arrangement design.

Figure 1 shows the novel EWOD platform system structure and the electrode design of the novel EWOD microfluidic chip by AutoCAD 2018 software, as shown in Figure 1a. The team defined the electrode pattern for the four basic operations of the EWOD technique and designed new channels for liquid beads to shuttle through the parallel and coplanar regions. The coplanar electrodes in the new EWOD platform are designed with interdigitated electrodes, divided into two groups of electrodes in the left and right directions, and both adopt a 13-finger design, as shown in Figure 1b. The working electrodes for the parallel-plate operating areas mainly include reservoir, generation, and square transporting electrodes.

This research used ITO glass as the dielectric wetting wafer substrate (Ruilong Optoelectronics Co. Ltd., Miaoli, Taiwan). ITO material has been widely used in optoelectronics due to its high transparency and optical properties, which are convenient for observation and operation under a microscope and have ideal conductive properties. The wafer fabrication process used the photolithography process. In the first step, we performed the standard cleaning. The second step involved spin coating the 1.2 µm photoresist with 3000 rpm for 30 s and developing the pattern. After hard baking with 120 °C for 5 min, we performed the third step, ITO etching. The chip is immersed to the FeCl_3_/HCl etching solution at 45 °C for 110 s. The fourth step involves coating the SU8-2002 negative photoresist (Kayaku Advanced Materials, Inc., Westborough, MA, USA) as a 1.5 µm dielectric layer. In the fifth step, we spin coated with Teflon AF1600 (Teflon™ AF, Wilmington, DE, USA) solution with a 1.2 µm thickness for the hydrophobic layer on the wafer. In the sixth step, we diced the wafer into an EWOD chip as shown in Figure 2.

### 2.2. Design and Fabrication Methods of Top Plate

In the droplet pull-out conjecture model proposed by the team, when a droplet moves forward, we considered the increase in the gap between the chip and the curved upper cover, as shown in Figure 3a. The primary resistance under the curved surface structure is the adhesion force of the droplet. The adhesion force causes the droplet to form a liquid bridge between the gaps even if the distance is increased under the parallel plate structure, which causes the liquid sample to quickly stick on the top plate. According to the previous research formulas, the adhesion force of droplets in the vertical direction can be quantified based on the droplet radius $R_{1}$ and $R_{2}$ dimensions and integrated with the contact angle [10,11], as shown in Figure 3b.

The droplet separates from the solid state, and the surface tension is an unignored technical point. The liquid adhesion is analyzed as shown in Equation (1).
(1)$$W_{adhesion}=F\cdot L=\gamma A$$
where γ is the surface coefficient per unit area of material, and A is the adhesion area of the droplet. Through a curvature surface design, the droplet pressure difference is described by the Young–Laplace equation at the adhesion model, as shown in Equation (2).
(2)$$W_{adhension}=\gamma_{lg}\left(\frac{1}{R_{2}}+\frac{1}{R_{1}}\right)\cdot \pi {R_{1}}^{2}\cdot \cos\theta_{Curve\;angle}$$
where $\gamma_{lg}$ is the tension at the interface between the liquids and gas. $\cos\theta_{Curve\;angle}$ is the relation among the contact angle at the curve substrate. Therefore, we use a dielectric wetting driving force to overcome the adhesion of liquid beads on the curved surface cover so that the liquid beads can be removed entirely from the top plate and pulled out from the capillary structure, as shown in the following Equation (3). The curved surface structure design is based on three surface structural properties: the droplet attachment area, curve angle, and capillary structure spacing.
(3)$$\begin{array}{l} \pi {R_{1}}^{2}\cdot \gamma_{lg}\left(\frac{1}{R_{2}}+\frac{1}{R_{1}}\right)\cdot \cos\theta_{curve\;angle}\\ \;\leq L\sin\varphi \left\{\cos\alpha \frac{\varepsilon_{0}\varepsilon V^{2}}{2h}-\gamma_{lg}\sin\alpha \left[\sin\theta_{\left(V\right)}-\sin\theta_{\left(0\right)}\right]\right\} \end{array}$$
where *L* is the length of the adhesion interface, φ is the contact angle when the droplet moves across to the next electrode, α is the wetting exponent, *h* is the distance of the parallel plates gap, and $\varepsilon_{0}$ and ε represent the dielectric constant and material dielectric constant, respectively. $\theta_{\left(0\right)}$ and $\theta_{\left(V\right)}$ represent the initial contact angle and contact angle with active voltage driving, respectively. The team used a mold process for top plate fabrication to create the top plate with a smooth curve surface. The design of the metal mold is used by SolidWorks 2020 software, as shown in Figure 4. A digitally controlled CNC milling machine machines the mold.

In this experiment, the team used silicon oil to create an oil environment to facilitate the generation and extraction of droplets. However, silicone oil can easily penetrate into the PDMS structure and cause the PDMS substrate to deform [12,13]. In order to solve the above problems, we selected Parylene C as the packaging material to isolate the contact between PDMS and silicone oil. We used ITO, the same as the dielectric wetting chip electrode material, as the ground electrode in the parallel plate operating area [14]. The curve top plate fabrication is shown in Figure 4b. First, we cleaned the metal mold by 95% alcohol. Second, we formed the PDMS structure with 10:1 A and B regent mixing, removed the bubble with vacuum environment, injected it into metal mold and let it solidify for 24 h. Third, we demolded the PDMS substrate and cleaned it with 95% alcohol. Fourth, we deposited the 5 µm Parylene C on the PDMS with a Polymer Deposition System (PDS) as the isolative layer. Fifth, we deposited the 230 nm ITO thin film by RF-sputter. Last, we sprayed the Teflon AF1600 solution on the top plate and placed it in a room temperature environment for 72 h to form the hydrophobic layer.

### 2.3. Experimental Buffer Preparation for System Testing

In this study, we selected DI water, propylene carbonate (PC, purchased from Echo Chemical Co., LTD., Miaoli County, Taiwan) and polyethylene glycol diacrylate (PEGDA, Merck Taipei, Taiwan) as the sample for novel EWOD platform performance testing.

For light-curing application research, we used PEGDA and lithium phenyl-2,4,6-trimethylbenzoylphosphinate (Li-TPO, purchased from Colorado Photopolymer Solutions, Boulder, CO, USA) light-curing powders as the primary test samples, and both were mixed with DI water to form aqueous solutions for research. The PEGDA solution has a concentration of 20%, and the Li-TPO light-curing powder solution has a concentration of 0.5%. The 20% PEGDA hydrogel material is a hydrophilic and biocompatible gel polymer material widely used in drug delivery tech and tissue engineering applications. At the same time, Li-TPO is a common photo-initiator that can generate free radicals when exposed to ultraviolet light [15,16].

The three-dimensional structure formed by PEGDA hydrogel can be classified according to the pore diameter and inter-pore structure, and it has different application fields according to the difference in structure size. Pores refer to the interconnected cavities within the hydrogel structure that allow the diffusion of solvents, gases, or other molecules; the inter-pore structure represents the scaffolding structure between polymer chains or networks within the hydrogel structure. Table 1 shows the comparison of different PEGDA structure properties and applications.

In this study, considering the versatile applications of hydrogel materials in the biomedical field and the two-dimensional (liquid) to three-dimensional (solid) transitions before and after ultraviolet (UV) light exposure, hydrogels were selected as the experimental samples to investigate the novel application of dielectric wetting. The aim is to synthesize poly(ethylene glycol) diacrylate (PEGDA) samples with different concentrations, resulting in variations in the size, structure, and swelling properties. The team intends to examine the system’s capability and stability in controlling sample volumes through this.

## 3. Performance Testing of Novel EWOD Platform

### 3.1. Electrowetting Performance of Novel EWOD Platform Coplanar Electrode

The experiment focuses on the electrowetting performance of coplanar electrodes on the chip by adjusting different operating signal voltages. The signal voltage range was gradually increased from 0 to 200 V, and different electrical signal frequencies were applied according to the different materials (DI water: 10 kHz, PC: 20 kHz, PEGDA aqueous solution: 10 kHz). The contact angle of a droplet is mainly recorded through a digital camera, and contact angle measurement is performed using Image-J 1.53t software. We used two general liquid materials, DI and PC solution, as functional testing samples of the chip to obtain condition verification for voltage operation.

According to the measurement result, the contact angle of the DI water sample dropped from 106.2° to 81.1°, with a total decrease of 25.1°, as shown in Figure 5. Compared to DI water, the contact angle of the PC sample on the Teflon hydrophobic layer is 73.9°. As the operating voltage increases, the contact angle of PC liquid beads gradually decreases to 52.9°, and the total decrease value is 21°, which is smaller than the total decrease value in DI water. The platform could change the contact angle significantly according to different operating voltages. The contact angle of the PEGDA aqueous solution on the surface of the Teflon water transfer layer is between DI water and PC material and gradually decreases as the PEGDA concentration increases. The contact angle of 5% PEGDA aqueous solution is 94.1°. As the voltage increases to 160 V, the contact angle can decrease to 60.5°. The contact angle of 10% PEGDA aqueous solution is 92.5° at 0 V and reaches 56.9° at 160 V. The contact angle of 20% PEGDA aqueous solution decreases from 86.8° to 50.1°. This shows that the uncured high-concentration hydrogel aqueous solution supplies more hydrophilic characteristics but is more likely to stick to the wafer surface and difficult to drive by the EWOD operation.

Through the electrowetting force equation shown below, the change in contact angle can be further converted into the driving force of the dielectric wetting system on the liquid sample. $\theta_{0}$ and $\theta_{v}$ are the contact angle values before and after the electrode is turned on, $\gamma_{sample}$ is the surface tension of the liquid sample (DI water is 72 mN/m, PC is 41.39 mN/m), and L is the length of the droplet normal to the force direction at the boundary between the two electrodes.
(4)$$F_{EW}=(\cos\theta_{0}-\cos\theta_{V})\times \gamma_{sample}\times L$$

Under different operating voltages, the maximum driving force that the new dielectric wetting system can exert on the DI water sample is 31.22 µN; for the PC sample, it is 13.38 µN. In the hydrogel sample test, the contact angle of the hydrogel decreases with increasing concentration and changes according to the cosine value of the contact angle, the magnitude of the driving force experienced by the hydrogel in the system is similar to that of DI water. This result is in line with the team’s expectations.

### 3.2. Droplet Volume Controlling the Performance of Novel EWOD Platform

In this section, the team conducted sub-droplet generation experiments and droplet pull-out operations with the different buffers mentioned before to verify the feasibility of the new EWOD system. In the sub-droplet generation test, droplet generation was conducted for six repeated experiments with different operating voltages to ensure the stability of the liquid bead generation.

According to the experimental test results, the sub-liquid bead areas generated by different samples are all located at 0.876 mm^2^, and the error rate between the liquid bead area and the electrode area is controlled within 1%. It will not change significantly as the voltage increases, as shown in Figure 6b. In the volume controlling test, we completed the manipulation process of the parallel-plate configuration system. We verified via platform stability that the droplet generation of the novel EWOD system has high precision and accuracy. This is the most critical part of the development of the EWOD system.

After successfully controlling the generation of droplets, we conducted the pull-out droplet test using the curved surface structure to pull droplets out of the parallel plate structure. There have been multiple operation processes and the switching sequence of the control electrodes to complete the droplet pull-out step, as shown in Figure 7.

We manipulated the system to generate a sub-droplet at step 1. In step 2, the first sub-droplet was transported to the first coplanar transport electrode. However, since the interval space between the top plate and bottom plate of the novel EWOD system increased as the droplet moved from the parallel-plate region to the coplanar-plate region with the curve structure, the droplet contact area decreased and became hard to cross to the next electrode. In order to make sure the droplet volume was large enough to cross the electrode, we repeated the creation operation and combined the sub-droplets. Step 4 was the second droplet creation. After combining the two sub-droplets, the droplet can cross the second coplanar electrodes but still cannot be pulled out from the parallel region in step 5. Step 6 was the third droplet creating operation. After combining three sub-droplets, the liquid bead could be pulled out from the parallel structure to the coplanar region, as shown in step 8.

During the droplet pull-out process, the generated sub-droplets ascend with the height of the curved surface structure. They are gradually stretched due to the combined influence of the adhesion force from the top cover and the driving force from the EWOD chip. This gradually reduces the contact area between the droplet and the top plate, making it challenging for the droplet to cross over to the next electrode. According to the analysis results from ImageJ, when the droplet reaches the position of the first coplanar electrode, the droplet’s area shrinks to 0.5 ± 0.03 mm^2^, which is only 60% of the original droplet area. It becomes difficult for the droplet to reach the second coplanar electrode, requiring the generation of a second droplet for accumulation. After the second droplet is overlaid, the area of the combined droplet reaches 0.823 ± 0.05 mm^2^, allowing it to reach the edge of the top cover, but it still cannot be fully extracted. Through the generation and combination of a third droplet, the area of the droplet on the second coplanar electrode reaches 1.214 ± 0.07 mm^2^. At this point, the droplet can be successfully pulled out of the curved structure and detached from the top cover of the parallel plate structure.

Through multiple experimental tests, the team successfully incorporated a fifth fundamental operation, “droplet pull-out”, into the original dielectric wetting system, which already included the basic droplet operations of creation, mixing, cutting, and transport. Under the current curved surface design and controlled electrode area conditions, the volume ratio between the generated droplet and the extracted droplet is 1:3. The volume of the extracted droplet is approximately 0.66 µL, ensuring no sample residue and complete extraction of the droplet.

For the multi-channel operation shown in Figure 8, we employed LabVIEW 2017 control software for droplet generation and pull-out operations. In the figure, steps 1~6, 7~12, 13~18, and 19~24 correspond to droplet generation and extraction operations in the left, right, lower, and upper curved channels, respectively. Take the left channel as an example; steps 1 and 2 were the first droplet creations in the left channel. Step 3 was the second droplet creation, and step 4 combined the first and second droplets. The third droplet creation was shown as step 5, and step 6 was the pull-out operation of the combined droplet of three sub-droplets.

### 3.3. Hydrogel Photocuring Application and Swelling Ratio Control Experiment

After verifying the droplet creation testing results and completing the droplet pull-out manipulation, we further demonstrated the system’s volume control performance for liquid samples and its ability to overcome the two-dimensional spatial constraints of parallel-plate structures by employing the novel dielectric wetting system in a hydrogel photocuring process.

Using a dual channel to generate hydrogel droplets and photopolymer droplets, the team overlaid different numbers of droplets, performed mixing, and exposed the mixture in an open area. The experimental workflow for hydrogel solution manipulation in the dielectric wetting system is illustrated in Figure 9. The experimental process first uses a pipette to drop appropriate volumes of hydrogel solution and photo-initiator solution sequentially from the holes on the curved surface onto the reservoir electrode of the EWOD chip. Subsequently, the sub-droplets of the hydrogel solution and the photoreceptor solution are generated, superimposed, and pulled out through electrode operation. After the sub-droplets are pulled-out from the parallel-plate region in the EWOD system, the droplets are mixed and exposed to UV light for curing in the single-plate region with a coplanar electrode array. Finally, during the exposure period, solidified samples were extracted using forceps and stored in a Petri dish with a diameter of 6 cm.

In this experiment, the volumes of the pull-out droplets have the volume of 0.662 µL (from three overlaid sub-droplets), 0.882 µL (from four overlaid sub-droplets), and 1.102 µL (from five overlaid sub-droplets), respectively. The exposure time was consistently set at 15 s. The photocuring experiments are analyzed using three sample combinations: 7.5%, 10% and 12.5% photo-initiator, as shown in Table 2. The experiments were conducted using a 20% PEGDA hydrogel, and the three mentioned formulations were organized into concentrations of 7.5%, 10% and 12.5% for data analysis and tabulation.

In Figure 10a, the final product dimensions after air-drying have an approximately 5% positive trend with increasing hydrogel concentrations. In addition to the radius size, the hydrogel’s water content and the polymer structure’s compactness are also affected. These two factors further contribute to the volume performance of the micro-scale hydrogel particles synthesized by the novel dielectric wetting system after natural air-drying shrinkage. Figure 10b shows that the air-dried samples were immersed in DI water for 24 h and measured by a digital microscope. This experiment demonstrated that the hydrogel samples do not show significant volume differences among different concentrations after swelling. This phenomenon is attributed to the excellent volume control capability of the novel dielectric wetting system in the presence of liquid.

We further discuss the swelling ratio to focus on the volume-controlling ability of the novel EWOD platform.
(5)$$Swelling\;ratio=\frac{{Volume}_{After\;swelling}}{{Volume}_{Before\;swelling}}$$

As shown in Figure 11, within the initial 1 min of immersion in DI water, the swelling volume ratio of the hydrogel samples gradually increased from 1.10 to 1.57. As the swelling time reached 2 min, the swelling ratio reached 1.95 and began to approach a plateau. Through image analysis, the hydrogel swelling volume fluctuated around 2.05.

The 10% sample group reached a steady swelling ratio of 1.90. Within the initial 1 min of immersion in DI water, the swelling volume ratio of the hydrogel samples gradually increased from 1.04 to 1.26. As the swelling time reached 2 min, the swelling ratio reached 1.42. Compared to the 7.5% sample group, the 10% sample group had not yet stopped the ascending swelling rate at this point. This sample group showed a noticeable plateau trend after 7 min.

In contrast, the 12.5% sample group had a relatively limited volume swelling ratio, reaching a steady swelling ratio of 1.45. Within the initial 1 min of immersion in DI water, the swelling volume of this sample gradually increased from 1.04 to 1.26 and reached 1.42 at 2 min, starting to plateau. In summary, the 7.5% hydrogel sample exhibited the most significant swelling ratio change, which was followed by the 10% sample. The 12.5% hydrogel showed the smallest swelling ratio with minimal volume changes during the water absorption process. We observed that the swelling rate of the 10% and 12.5% samples had a large standard deviation in the first 150 s. During this time, the trend of swelling efficacy is mainly influenced by the surface area and environmental evaporation, both of which affect the contact for moisture absorption. As time progresses, the moisture evaporation and absorption within PEGDA reach equilibrium, allowing us to clearly observe different swelling rate performances among PEGDA concentrations.

According to the research results on the swelling ratio of hydrogels over time, the 7.5% sample group exhibited a higher volume swelling ratio of 2.05. This indicates that the PEGDA hydrogel has a higher water capacity with low hydrogel concentration. The experiment result is similar to the radius size measurement.

According to the previous research results by Jae-Woo Lee, increasing the molecular weight of PEGDA can promote an increase in the pore size and swelling ratio of hydrogels. However, under the same molecular weight conditions, increasing the weight ratio of PEGDA can enhance the cross-linking density of hydrogels and reduce the swelling ratio [24]. Through this application validation, the volume control capability of this platform demonstrates excellent liquid adsorption efficiency in PEGDA synthesis, including the size differences, water capacities, and swelling speeds of different hydrogel concentrations.

## 4. Conclusions

This study successfully implemented a dielectric wetting system with a curved surface structure capable of freely navigating between double-panel and single-panel configurations. The team used DI water, PC and hydrogel PEGDA solution as the test droplet materials in the validation experiment. In the droplet generation tests, we could generate droplets on a PDMS substrate, and the differences in the generated droplet areas were within 2%. The precise droplet control was achieved regardless of the droplet material type. Based on the current electrode dimensions and curved surface angle structures, it was determined that three droplets needed to be generated to meet the volume requirements for hydrogel droplet pulling out. Under stable system operations, we achieved the feasibility of the novel dielectric wetting system in volume control and three-dimensional particle synthesis through experiments controlling the hydrogel swelling ratio.

The novel EWOD platform represents a significant advancement in the field of microfluidics. While it offers clear benefits over traditional systems, its unique design and operational principles make direct comparisons difficult. Instead, this platform should be evaluated based on its ability to meet the specific needs of modern applications particularly in biomedical diagnostics and advanced drug discovery technologies. This novel approach holds the potential to set new standards in the industry, providing a versatile and efficient solution for a wide range of applications.

## Figures and Tables

**Figure 1 micromachines-15-00797-f001:**
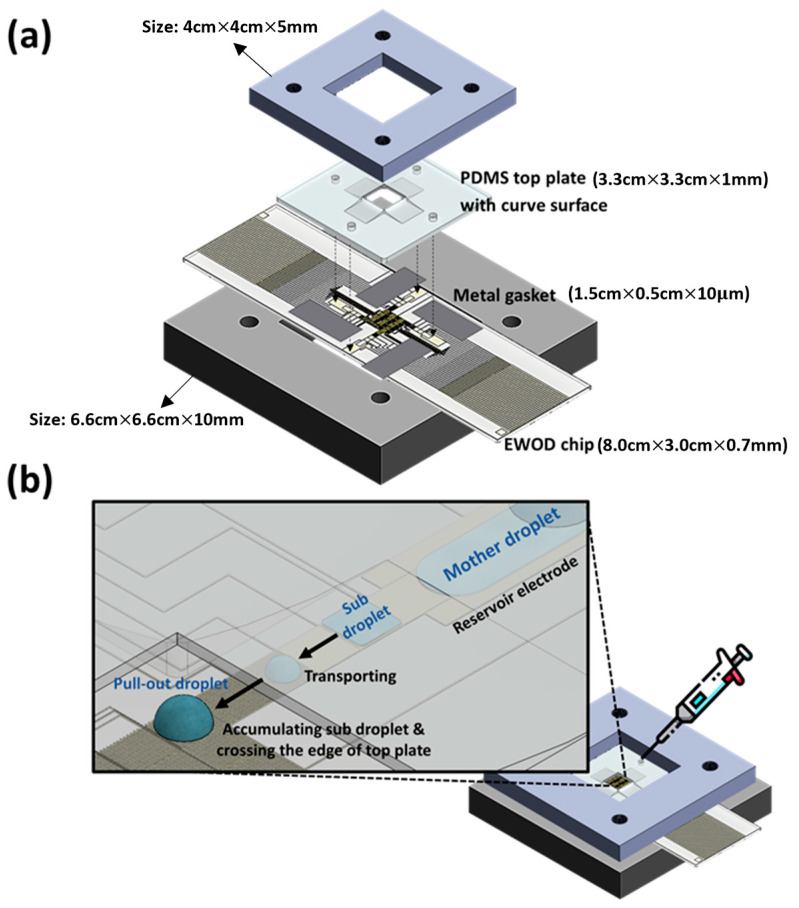
(**a**) Combination of the novel EWOD microfluidic platform. (**b**) Schematic diagram of droplet-pulling operation with curve PDMS top plate. In the novel EWOD platform, the team uses the metal gasket to create the interval space of the EWOD system and design the fixture to stabilize the novel EWOD system in the sandwich structure.

**Figure 2 micromachines-15-00797-f002:**
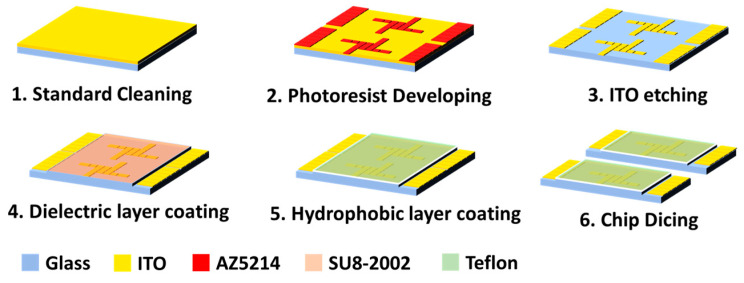
Fabrication process of EWOD chip.

**Figure 3 micromachines-15-00797-f003:**
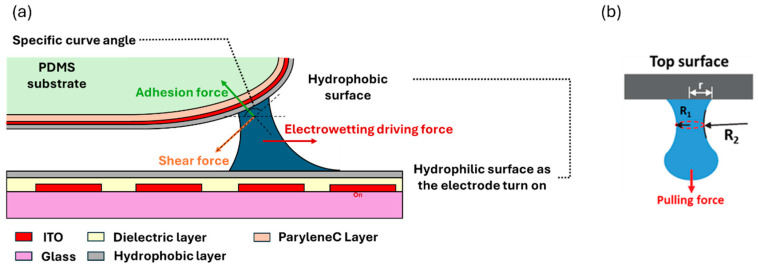
(**a**) Schematic diagram of forces acting on the droplet at the boundary between the parallel plate and single plate. (**b**) Schematic diagram of droplet extension and pulling force situation for fluid adhesion. *R*_1_ and *R*_2_ are the radii of curvature, and r is the contact radius.

**Figure 4 micromachines-15-00797-f004:**
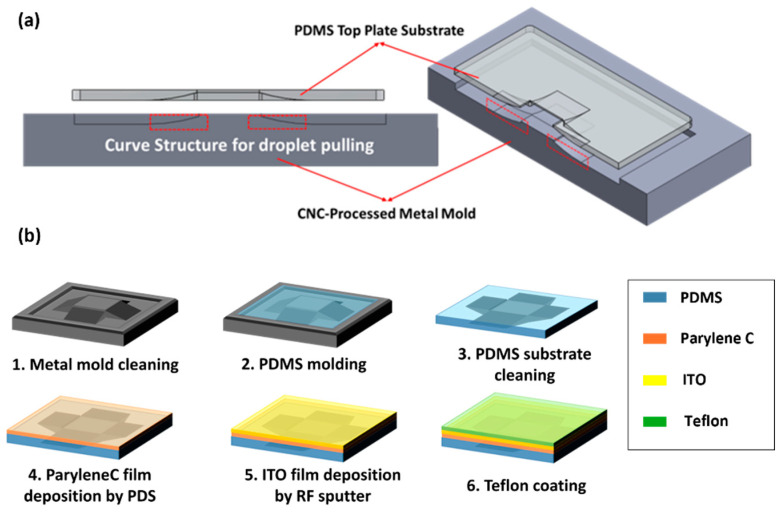
(**a**) Schematic diagram of metal mold and curved top plate molding process. (**b**) The curve structure top plate fabrication.

**Figure 5 micromachines-15-00797-f005:**
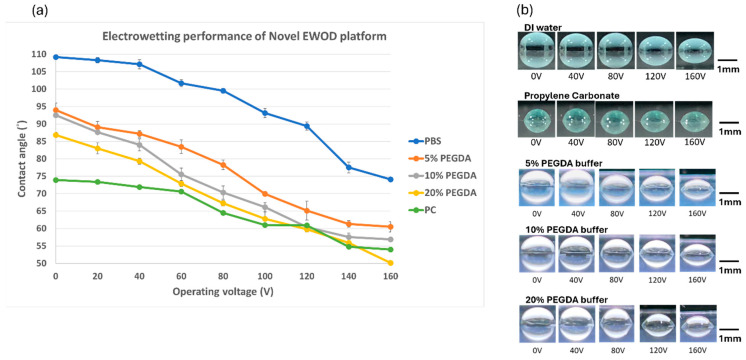
(**a**) Electrowetting performance of the coplanar electrode in the EWOD system. (**b**) Different material droplet optical images of the contact angle result. The experiment was run in quintuplicate.

**Figure 6 micromachines-15-00797-f006:**
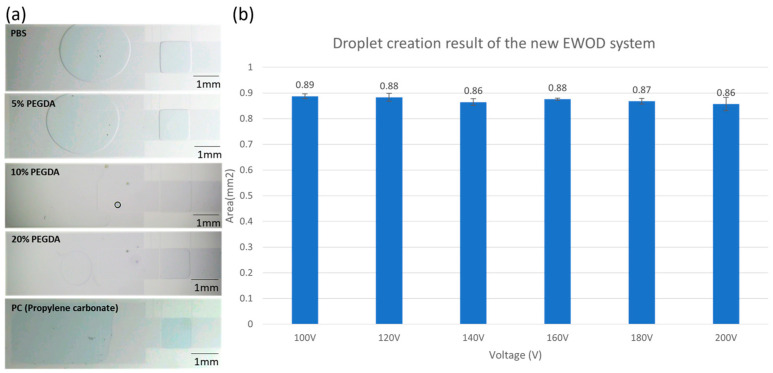
The droplet volume controlling performance. The experiment was run in quintuplicate. (**a**) Demonstrate the sub-droplet of different sample generations at 100 V electric signal. (**b**) Experimental results of sub-droplet generation test under different voltages.

**Figure 7 micromachines-15-00797-f007:**
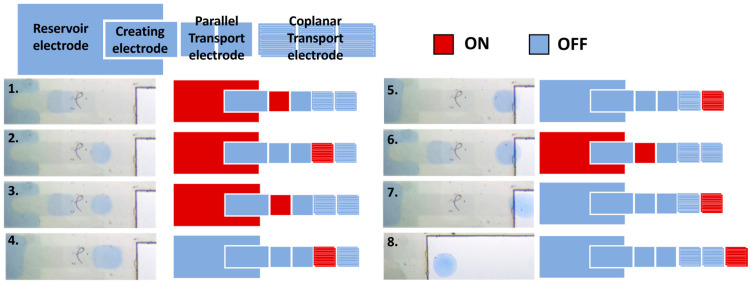
Droplet pull-out operation and the switching sequence of electrodes. The process is divided into 2 parts corresponding to the real-time image and signal control illustration.

**Figure 8 micromachines-15-00797-f008:**
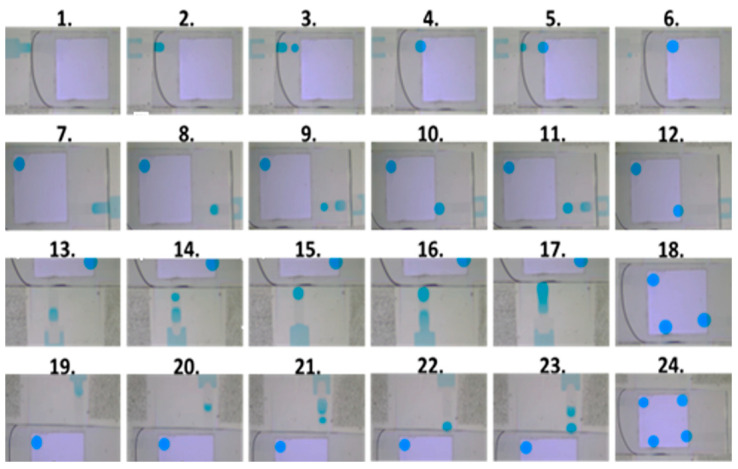
The operation process of multi-channel droplet generation and pull-out.

**Figure 9 micromachines-15-00797-f009:**
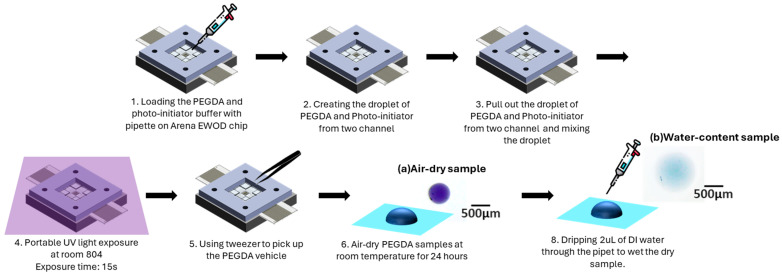
The experiment process of hydrogel photocuring application (**a**) Naturally air-dried PEGDA microparticles before swelling in step 6. (**b**) Naturally air-dried PEGDA microparticles after swelling in step 7.

**Figure 10 micromachines-15-00797-f010:**
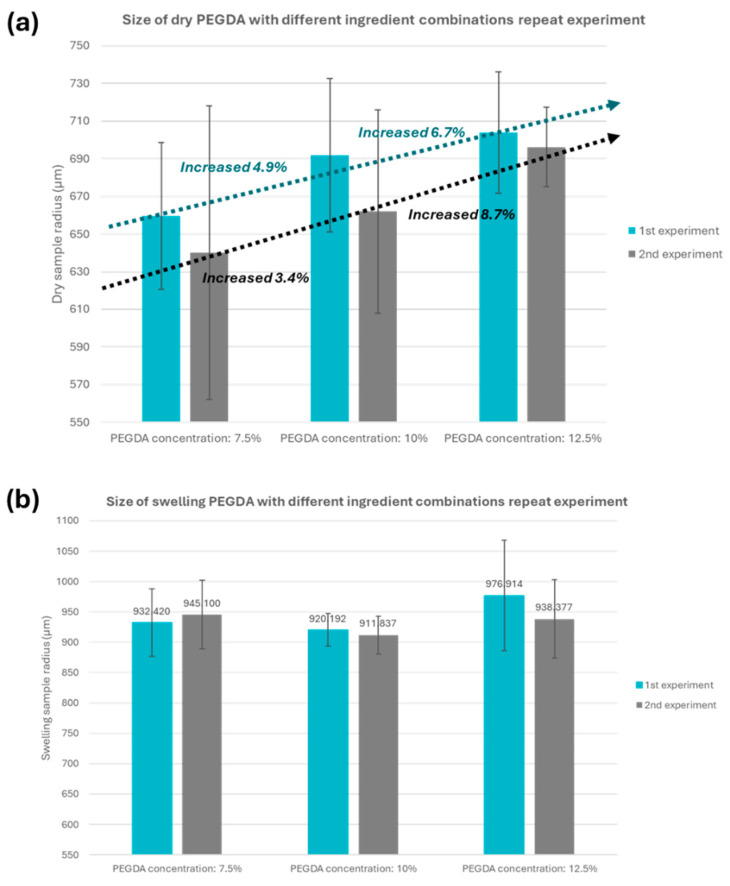
Size measurement results of air-dried and swelling PEGDA samples. (**a**) Radius size (in µm) of naturally air-dried PEGDA microparticles and (**b**) swelling PEGDA microparticles. Since different hydrogel and photocuring agent liquid concentration combinations have different structural densities of the synthesized hydrogel particles, the structural stability, swelling performance, water content and other physical properties of the gel during the swelling process are different.

**Figure 11 micromachines-15-00797-f011:**
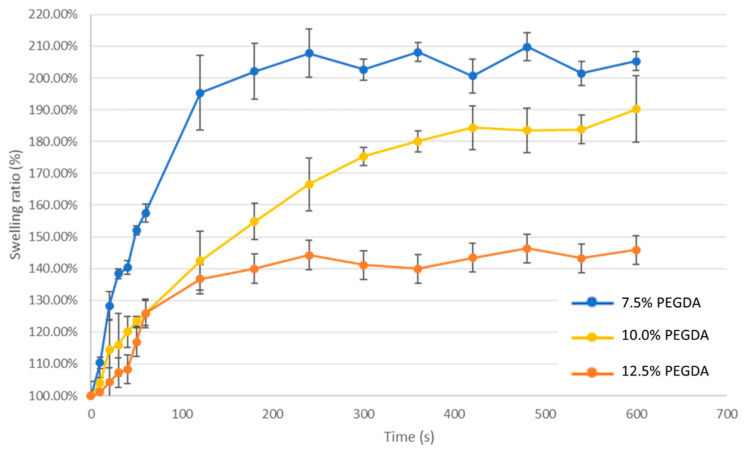
Chart of swelling ratio growth trend of hydrogel samples at various ratios.

**Table 1 micromachines-15-00797-t001:** PEGDA structure and application comparison table.

Type	Porous Size	Properties	Application
Macroporous	10~1000 µm	Liquid can flow easily with the porous size of this group, which is conducive to the penetration and diffusion of required nutrients and oxygen and conducive to the removal of waste products produced by cells [17,18].	Provide an environment conducive to the growth of tissues such as bones, articular cartilage, skin, and blood vessels for research on cell adhesion, proliferation, and tissue regeneration [19].
Mesoporous	10~100 nm	High surface area and pore volume enable the precise control of molecular diffusion and interactions within the hydrogel matrix [20].	Drug package-related research, including medicine payload and precise releasing [21].
Microporous	<2 nm	It exhibits an exceptionally extremely high surface area ratio and can absorb large amounts of liquid. The presence of micropores also enhances the mechanical properties of hydrogel material, including compressibility and flexibility.	The hydrogel structure of this type has specific performance for applications such as wound dressings, tissue engineering scaffolds and biosensors [22,23].

**Table 2 micromachines-15-00797-t002:** Photocuring experiments sample combinations comparison information.

Hydrogel Concentration	Volume of 20% PEGDA	Volume of Photo-Initiator
7.5%	0.662 µL (3 sub-droplets)	1.102 µL (5 sub-droplets)
10%	0.882 µL (4 sub-droplets)	0.882 µL (4 sub-droplets)
12.5%	1.102 µL (5 sub-droplets)	0.662 µL (3 sub-droplets)

## Data Availability

The original contributions presented in the study are included in the article, further inquiries can be directed to the corresponding author.

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
