# Peer review of "A Novel EWOD Platform for Freely Transporting Droplets in Double and Single-Plate Structures"

_micromachines, 2024, doi:10.3390/mi15060797_

Round 1

Reviewer 1 Report

Comments and Suggestions for Authors

The current paper by Nuoh Chang et al. focuses on a new type of dielectric wetting microfluidic operation platform, which combines a parallel plate region and a single plate region with a curved surface structure as the connection structure, and is tested with a few liquid materials such as DI water, propylene carbonate (PC), and PEGDA to assess its performance in volume control and liquid sample manipulation.

In my opinion, the research could provide a valuable addition to the field, and I recommend its publication once certain updates are made. My comments are below.

1- The authors need to improve the language to enhance readability and ensure better understanding by readers.

2- The introduction should better highlight the paper's main points, emphasizing the advantages of the new platform and addressing what is lacking in the field that this paper can improve.

3- Comparisons of the performance of this new system against standards, and other related studies from the literature are required to improve the significance of the work. For example, this could be a new table, comparing literature results with this new one.

4- The authors should include supplementary information to provide additional details and expand on the fabrication process, particularly more information about how they coat indium tin oxide (ITO) on Parylene C and the stability of their fabrication.

5- The stability of the platform needs to be evaluated and explained in the paper.

6- The authors performed experiment on only one curve angle. I recommend including a plot that shows the relationship between actuation voltage and various curve angles to better understand how the curve angle impacts the voltage.

7- In order to improve clarity for readers, the authors should add the schematic diagram of forces acting on the droplet at the boundary between parallel plate and single plate.

8- In conclusion section, the authors need to amplify how this research work contributes to forwarding the field of study.

Comments on the Quality of English Language

The authors need to improve the language to enhance readability and ensure better understanding by readers.

Reviewer 2 Report

Comments and Suggestions for Authors

In this work, Yii-Nuoh Chang et al developed a new type of dielectric wetting microfluidic operation platform to realize the manipulation of droplet including splitting, transporting and mixing in different region of the EWOD system. In general, I think the work is suitable to publish on the journal of micromachines. However, some problems should be addressed before acceptance.  

1. As described by the authors, the EWOD system is divided into PDMS substrate top plate and ITO glass substrate EWOD chip. It is recommended to describe the detail sizes of individual parts and exhibit in the Figure 1 and Figure 3.

2. In the experimental section, some experimental conditions are not explicitly indicated, i.e. the fabrication process of EWOD chip.

3. It is known for that the thickness of the hydrophobic layer would affect the electrowetting performance of EWOD system. Therefore, the authors should point out the thickness of the AF1600.

4. In the text, Parylene C was selected as the packaging material to isolate PDMS and silicon oil. I want to know how to realize the adhesion of Parylene C and PDMS. In addition, Parylene C should be showed in Figure 2.

5. In the droplet pull-out process, sub-droplets were stretched under the combined effect of adhesion force from the top cover and the driving force from EWOD chip. Do the curve structure shown in Figure 4 influence the adhesion force? And the relationship between them is recommended to point out.

6. The optical image of the droplet contact angle in different operation voltage should be insert in Figure 5. Why the frequencies of voltage respectively applied in DI water, PC, PEGDA are different? Moreover, the error bar is recommended to add in Figure 5 for indicating the repeatability and stability of the EWOD device.

7. In Figure 11, the swelling ratio of PEGDA decreased with the mass weight of PEGDA within ~75 s. Whereas the increasing rate of swell ration of 10% PEGDA is obvious bigger than that of 12.5% PEGDA. I think the author should discuss this phenomenon and put forward the reason.

Comments on the Quality of English Language

1. The English should be improved.

2. The resolution of the figures in this work should be improved. The name of dependent variable in Figure 6 is indeed unclearly.

3. The letters in every equation should be explained for the representation.

4. The abbrevition should be appear on the first occurrence.

Round 2

Reviewer 1 Report

Comments and Suggestions for Authors

The authors considered my comments by providing more detailed explanations of the fabrication process and incorporating schematic diagrams that illustrate the forces acting on the droplet at the boundary between the parallel and single plates. Furthermore, in the conclusion section, they more clearly clarified how this research contributes to advancing the field of study. However, I recommend that the authors add further details to the fabrication description, specifically a better explanation of the ITO etching process in the Materials and Methods section.

With these improvements, I would recommend the revised version for publication.

Author Response

Thank you for your recommendations. We have added the etching detail of fabrication in line 91(red color).